# Sequence Analysis of Ancient River Blocking Events in SE Tibetan Plateau Using Multidisciplinary Approaches

Yiwei Zhang [1], Jianping Chen [1], Qing Wang [1,*], Yongchao Li [2,3,4], Shengyuan Song [1], Feifan Gu [1] and Chen Cao [1]

1   College of Construction Engineering, Jilin University, Changchun 130026, China; yiweiz18@mails.jlu.edu.cn (Y.Z.); chenjp@jlu.edu.cn (J.C.); songshengyuan@jlu.edu.cn (S.S.); guff@mails.jlu.edu.cn (F.G.); ccao@mails.jlu.edu.cn (C.C.)
2   Key Laboratory of Shale Gas and Geoengineering, Institute of Geology and Geophysics, Chinese Academy of Sciences, Beijing 100864, China; liyongchao@mail.iggcas.ac.cn
3   University of Chinese Academy of Sciences, Beijing 101408, China
4   Innovation Academy for Earth Science, Chinese Academy of Sciences, Beijing 100864, China
*   Correspondence: wangqing@jlu.edu.cn; Tel.: +86-13504321568

**Abstract:** The temporary or permanent river blocking event caused by mass movement usually occurs on steep terrain. With the increase of mountain population and land use pressure and the construction of water conservancy and hydropower projects, river blocking events have gradually attracted people's attention and understanding. The area in this study is affected by strong tectonic activity in the Jinsha River suture zone and the rapid uplift of the Tibetan Plateau. In the past 6000 years, there have been at least five obvious river blocking events in the reach. The number and density are very rare. Combining field investigation, indoor interpretation, laboratory tests, optically stimulated luminescence (OSL) dating, SBAS-InSAR and previous studies, multidisciplinary approaches are used to systematically summarize the analysis methods and further the understanding of one river blocking event and multiple river blocking events from different perspectives. Especially in multiple river blocking events, we can get the wrong results if interaction is not considered. Through this study, the general method of analyzing the river blocking event and the problems that should be paid attention to in sampling are given, and relatively reliable historical results of river blocking events are obtained. This method has applicability to the identification and analysis of river blocking events and age determination of dams with multiple river blockages.

**Keywords:** river blocking event; landslide dam; multidisciplinary approaches; dating and SBAS-InSAR





## 1. Introduction

The Qinghai–Tibet Plateau is the highest plateau in the world, and it is still increasing [1–3]. The upper Jinsha River is located on the southeast margin of the Qinghai–Tibet Plateau, significantly affected by tectonic uplift. There are steep slopes, deep valleys, and widely developed mass movement [4]. When geomorphic processes such as landslides cut off the river, they lead to the formation of temporary or permanent stream blockage [5,6] and present the greatest threat to people and property [7,8]. A recent blockage of the Jinsha River occurred on 10 October 2018, and the Baige landslide river blocking event (31°4′51″ N; 98°43′01″ E) was about 200 km upstream of our study area. Then, secondary hazards can be induced when landslide dams form and fail, including dam breach, upstream inundation, and downstream flooding [9]. Jinsha River is one of the rivers with the largest water resource potential in China [10], so it is a relevant area in the development of water conservancy and hydropower projects. In addition, it is of great significance to carry out geological hazard research on this section [11,12].

As a complex geological and geomorphic phenomenon, especially in ancient times, the blocking of Jinsha River provides a lot of information for its development history, quaternary seismic activity, formation and evolution of catastrophic geological disasters

in large valleys, and better understanding of the geological environment caused by earthquakes [13]. In the research area of about 26 km, there are several large landslide bodies, six of which could be identified as river blocking events, named Wangdalong I and II (WDL I and II), Rongcharong (RCR), Suwalong (SWL), Suoduoxi (SDX), Gangda (GD). Geologists are interested in learning ancient information from these events.

In recent years, there have been numerous studies conducted concerned about dammed lakes in the Quaternary, which developed on the major rivers of the Qinghai–Tibet Plateau [2,13–17]. The research methods are becoming more quantitative and systematic, but there are still problems such as the lack of details and theoretical support due to limited information obtainable from Quaternary sediments, which makes it possible to ignore certain information about the river blocking event, and the analysis results are biased, especially in the case of multiple river blocking events in one river section.

Therefore, based on previous studies, we combined field investigation, experimental data and our own analysis methods, hoping to play a complementary role in the research of river blocking events. The goals of this study include: (1) characterizing the geomorphological and sedimentological features of the ancient river blocking events; (2) determining the formation and breaching time of ancient river blocking events in order to contribute to the analysis of paleoclimatic and paleotectonic activity; (3) summarizing key ideas about analysis of multiple river blockages; (4) summarizing of general flow and precautions of river blocking research, providing some further insights in prospective research analysis, experiments, etc.

## 2. Study Area

### 2.1. Regional Geologic Setting

The study area is located on the upper reaches of Jinsha River in the southeastern margin of Qinghai–Tibet Plateau (Figure 1a). Tectonically, the structure of the study area is strong and there are many deep faults around (Figure 1b). There are two main groups of active faults in the study area: Yangla–Dongzhulin Fault zone ($F_3$) and Zeng Datong North–South fault ($F_4$), which are still active with the estimation of strike slip rates to be 6~7 mm/a and vertical rates to be 2~3 mm/a [18]. The study reach belongs to arid or semiarid climate [3], resulting in serious weathering and poor vegetation development on both sides of the bank slope. Along both sides of the valley, the exposed rocks are mainly schist, granodiorite, marble, limestone and granite (Figure 1c). These conditions lead to the development of jointed fractures in rock mass.

### 2.2. Geologic Setting of Each Dam

Affected by tectonic activities, there are faults near the dam body in this area, which make the surrounding rock mass more broken and joint fissures to develop, and this is the main reason for the existence of landslide dams in this river section (Figure 2). Except for GD, there are several kinds of lithology around the dam body, and different lithology boundaries are places with poor mechanical properties. The inclined plate rock mass around the dam body is prone to bending and cracking. These are unfavorable factors for the stability of the riverbank slope.

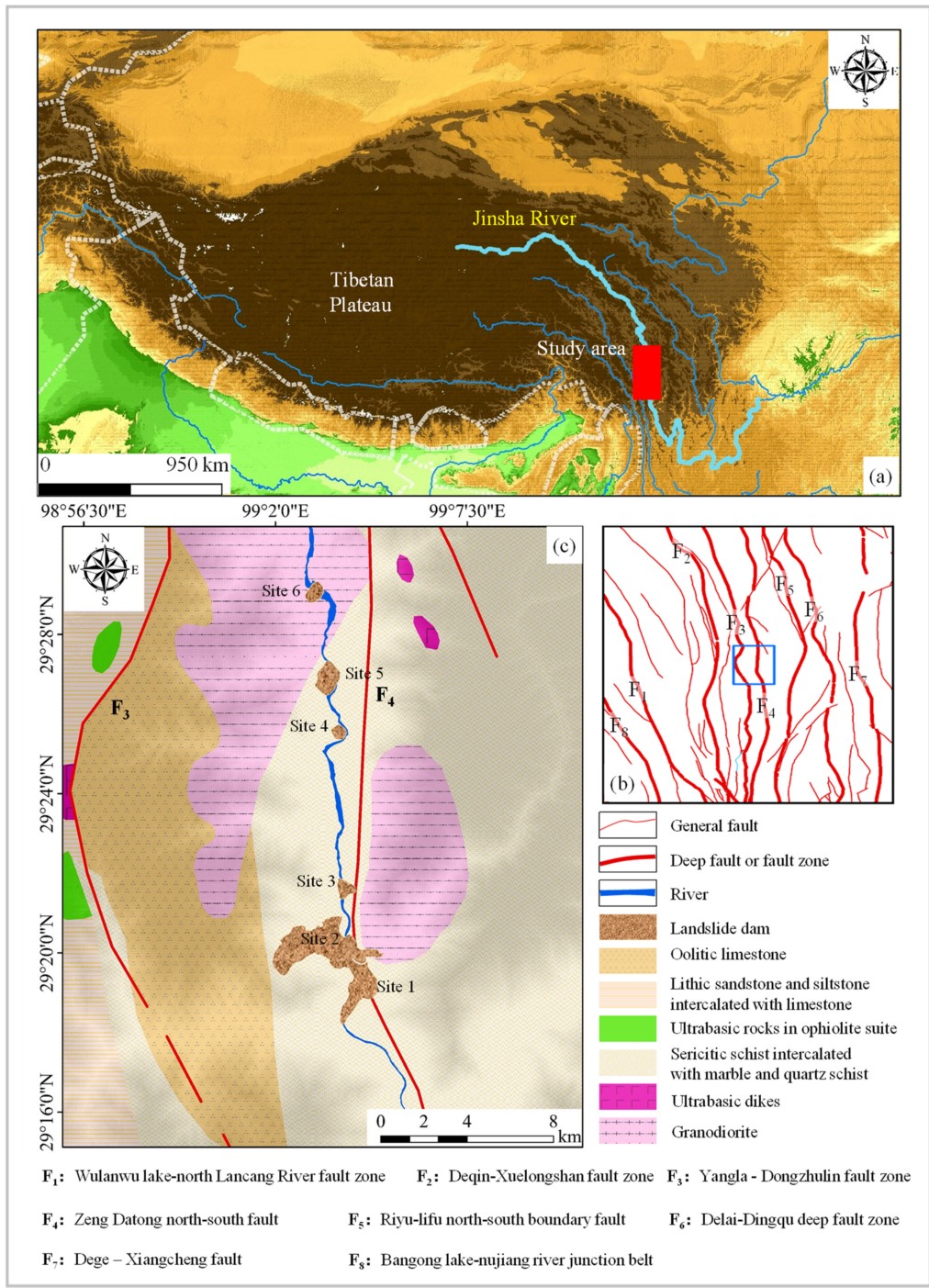

**Figure 1.** Map showing the locations and regional geologic settings (**a**) located on the southeastern edge of the Tibetan Plateau, (**b**) tectonic outline map (according to 1:1,000,000 geological map), (**c**) regional geological map (according to 1:1,000,000 geological map).

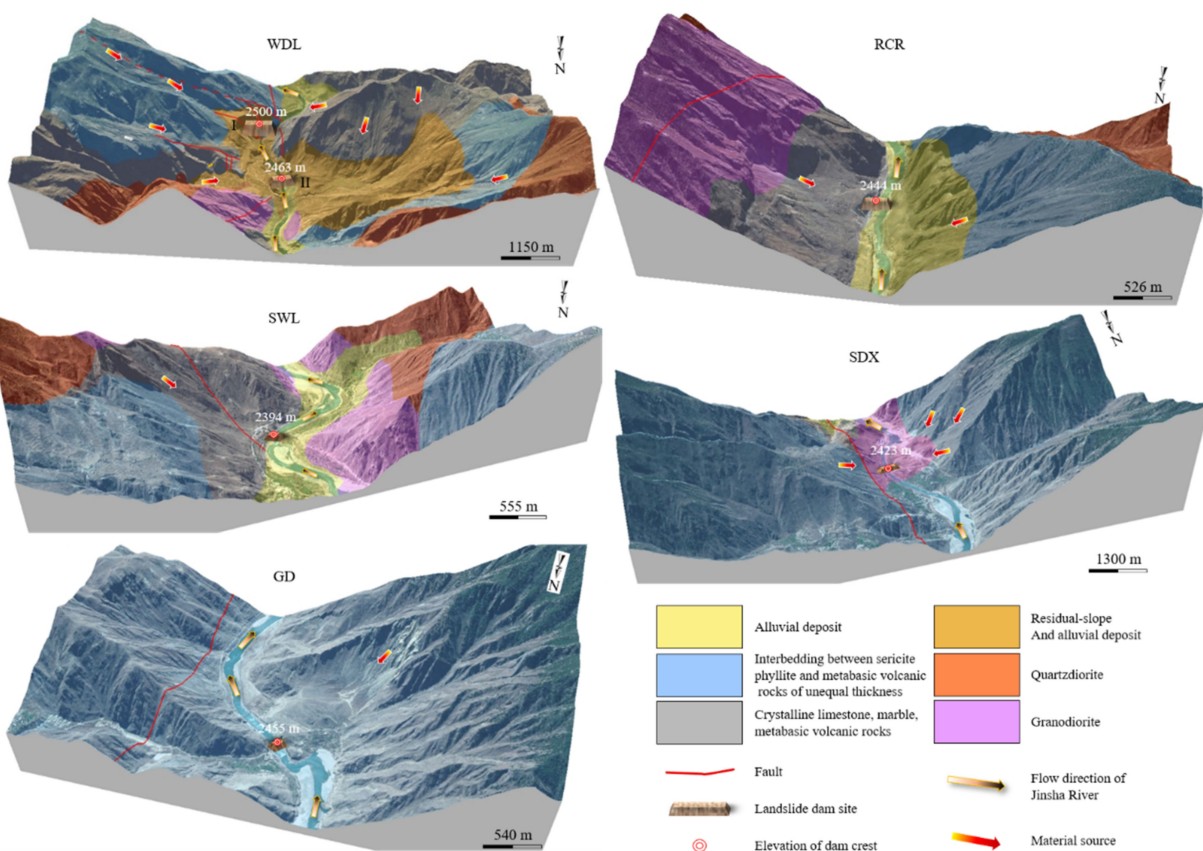

**Figure 2.** Map showing the specific geological settings of the dam site. Note: Due to the limitations in data collection, WDL, RCR and SWL were generated according to 1:50,000 geological map; SDX and GD were generated according to 1:200,000 geological map. (Note: these figures were created by ArcGIS 10.2).

## 3. Methodology

Uniformitarianism, also called the comparative–historical method, is an important paradigm in the process of geological research [19,20]. We can deduce the conditions, processes and characteristics of ancient geological events by using the existing laws of geological action, through the geological phenomena and results left over by various geological events. 'The Present is the key to the Past' is the uniformitarian paradigm [19,21] and nature is the best geological museum and laboratory, so field investigation is the premise and foundation of geological research and it is the traditional thinking method of geology. Therefore, a uniformitarian approach with the combination of modern science and technology is applied to improve the analysis of river blocking events.

For an ancient river blocking event, there are many pertinent research methods. A method called the "trinity" combination of residual landslide dams, upstream lacustrine sediments and downstream break-outburst sediments has been proposed [2,22]. In short, the fact that the river is blocked has basically reached a consensus and the method is reliably suitable for landslide dams as classified by Costa and Schuster [23].

### 3.1. Analysis of an Independent Landslide Damming of River Event

For one landslide damming river event, some traces would be left near the location where the river was blocked. Considering the fact that there would be less key direct information left in some river blocking events, it is necessary to analyze and summarize from different dimensions and perspectives.

### 3.1.1. Evidence from Remote Sensing Interpretation

The occurrence of the river blocking event requires the joint action of two aspects. The first is the stream channel, the other is blocking dam. For the remaining Quaternary ancient river blocking event, the square is generally large. Therefore, we can search along the river to find the landslide dam body on both sides of the river using remote sensing images (Figure 3), which is the potential evidence of blocking the river. Therefore, there will be greater changes in topography, including the change of bank slope morphology and the phenomenon of river diversion. In Google Earth, we can roughly circle the scope of the remaining dam block to find the source of blocking material. At the same time, we can also use DEM data for 3D modeling in GIS software such as ArcGIS to obtain some relative geometric parameters, such as the accumulation area, the accumulation length along the river, the accumulation width perpendicular to the river direction and the accumulation thickness of the dam body at the collapse.

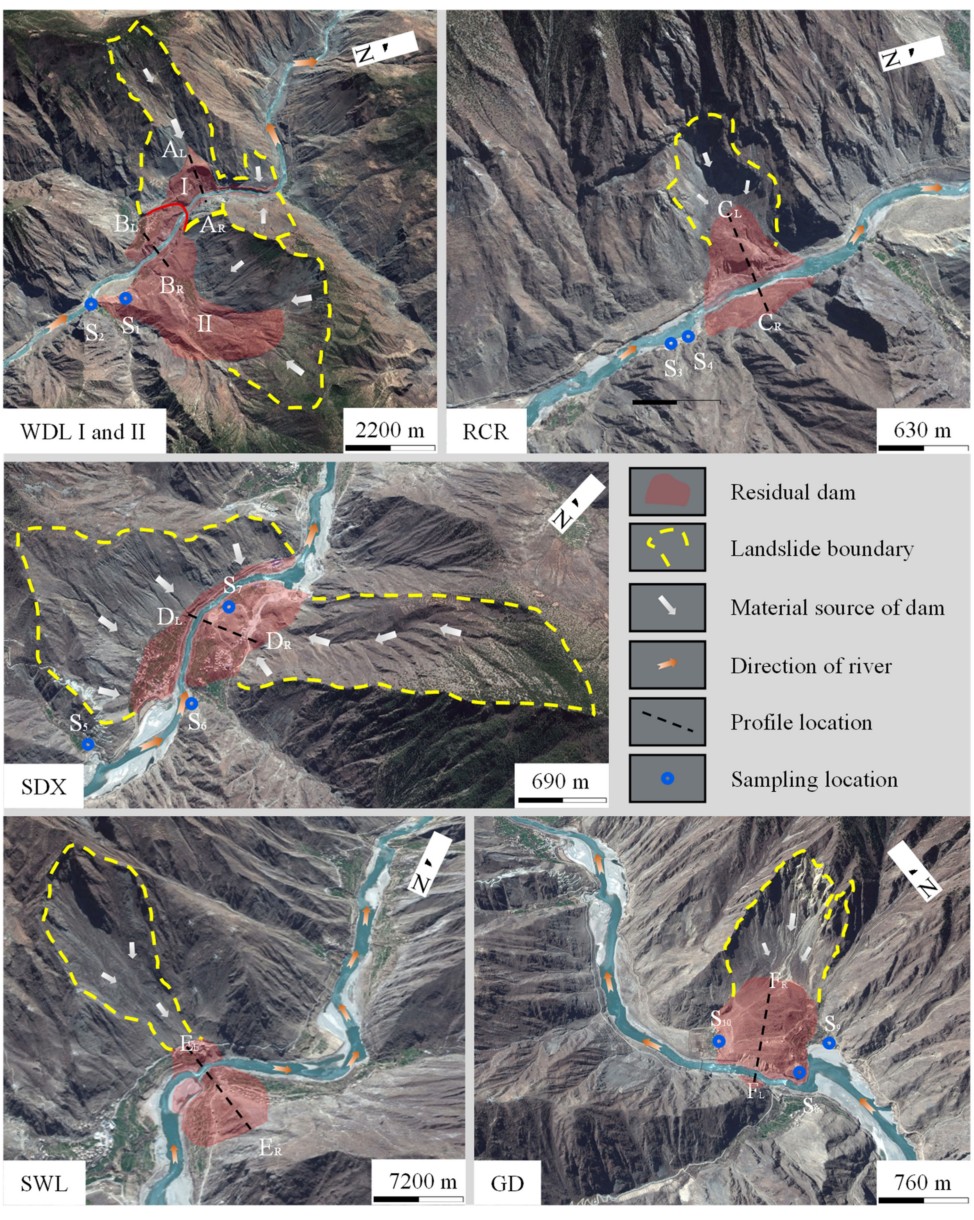

**Figure 3.** Remote sensing map of river blocking event. (Note: these images are from Google Earth).

### 3.1.2. Evidence from Morphology

After determining the approximate location of river blocking, the phenomenon of river blocking can be more accurately identified by detailed field investigation. Taking on-site photos of GD as an example (Figure 4), we can see that the dam bodies on both sides of the Jinsha River have geometric continuity and a good curve can be obviously observed by abstracting it into geometry. In the photographs taken, the geometric size of the river blocking body is well identified and recognized. At this time, important information such as the maximum thickness, average thickness of the accumulation body, and the geometric position and size of the breach are recorded, which are of great significance for the verification of the inversion results of the numerical simulation.

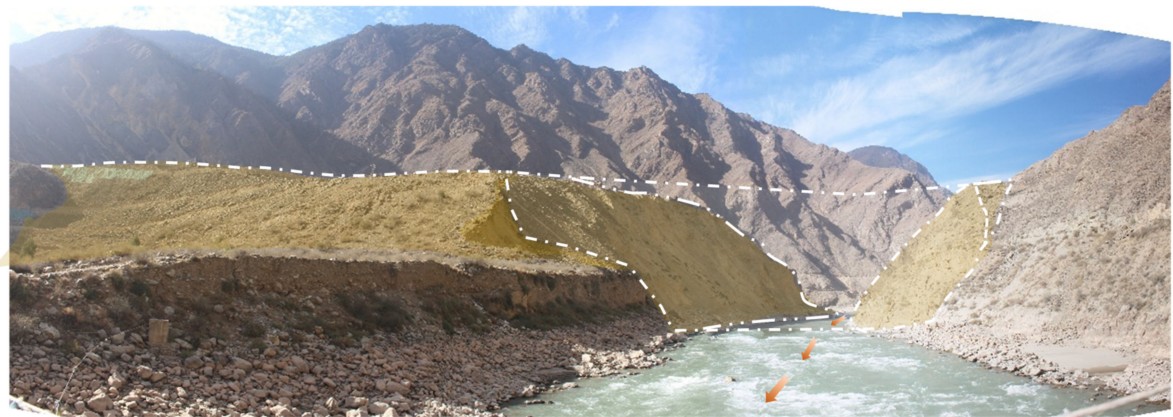

**Figure 4.** Geometry of GD landslide dam.

### 3.1.3. Evidence from Geology

For a river blocking dam where we can find a source area, geological continuity usually is maintained. That is, the lithology of the dam is consistent with the lithology of the material source area. Similarly, we can look for material sources on this basis. When the lithology of the bank slopes on both sides is inconsistent, and the lithology of the residual dam on both sides is consistent, we can infer the occurrence of river blockage and determine the source of material. However, in turn, when both sides of the bank slope have the same lithology, even if the internal lithology of the residual dam is consistent, it cannot be inferred that the material comes from one bank, which requires further analysis combined with remote sensing interpretation.

### 3.1.4. Evidence from Sedimentology

In the study section, there are a lot of fine-grained sediments (Figure 5). These lacustrine sediments not only directly reflect the sedimentary environment, but also reflect the hydrodynamic conditions of the transport medium. In order to determine the grain size characteristics of the lake sediments, the samples were taken from the lake sediments during the field investigation. As the lacustrine sediments particles are small, they can be all brought back for grain size analysis to obtain cumulative curve of particle size. In our laboratories, we used a hydrostatic sedimentation experiment to measure the grain size. Then we can get particle size characteristic parameters (Table 1). Firstly, according to Eli law, the diameter of the bed load moving on the riverbed is proportional to the square of the flow velocity (Equation (1)). In this study, we selected the maximum $d_{50}$ value in Table 1 as the calculation data. By the following assignment, $d = d_{50max} = 2.6 \times 10^{-5}$, $r_S = 2700$, $r = 1000$, $g = 9.8$, we calculated $V = 1.73 \times 10^{-2}$ m/s, which is far less than the normal velocity of Jinsha River. Secondly, according to the Stokes formula, similarly, 0.026 mm is chosen as the calculation particle size, and the average temperature of Jinsha River is selected as 9.2 °C (according to the Batang Hydrological Station) to select the particle size calculation coefficient. The sedimentation velocity $v = 8.146 \times 10^{-7}$ m /s is calculated and

the setting time is about 568 days when the settling height is 40 m. Through the above rough calculation, it is concluded that a certain thickness of fine sediment layer on the upstream of landslide body must be formed in a stable still environment where the river is blocked. Therefore, the existence of the lacustrine deposits layer can effectively reveal the river blocking event.

$$d = \frac{rk}{2gf(r_s - r)} \cdot V^2 \tag{1}$$

where $V$ is the velocity acting on the surface of sediment particles, m/s; $d$ is the diameter of sediment particles; m$r_S$ is the density of sediment particles, kg/m$^3$; $r$ is the density of water, kg/cm$^3$; f is the coefficient of friction; g is the acceleration of gravity, 9.8 m/s$^2$.

$$v = \frac{2}{9} \cdot \frac{(\rho_S - \rho_W)g}{\eta} \cdot r^2 \tag{2}$$

where $v$ is the sedimentation velocity of soil particles, cm/s; $r$ is the radius of soil particles, cm; $\rho_S$ is the density of solid particles, g/cm$^3$; $\rho_w$ is the density of water, g/cm$^3$; $\eta$ is the coefficient of dynamic viscosity of water, Pa·s; g is the acceleration of gravity, 980 cm/s$^2$.

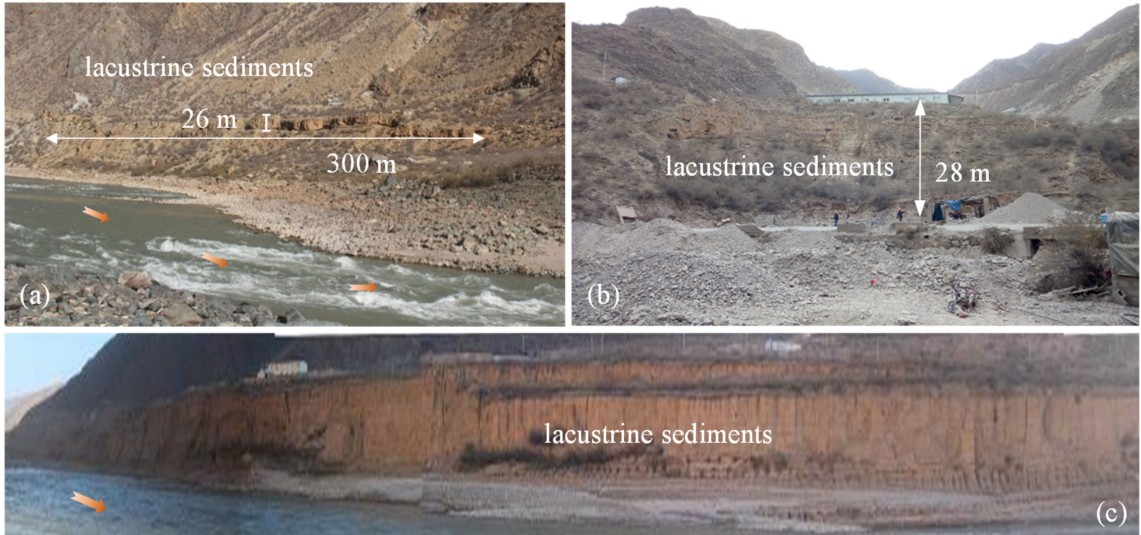

**Figure 5.** Lake sediments formed in dammed lakes. (**a**) In WDL-RCR reach; (**b**) in WDL-RCR reach; (**c**) in RCR-SWL reach.

**Table 1.** Characteristic parameters of cumulative percentage curves of lacustrine sediments.

| Sample | Effective Size | Mean Size | Control Size | $d_{30}$/mm |
|:---:|:---:|:---:|:---:|:---:|
| | $d_{10}$/mm | $d_{50}$/mm | $d_{60}$/mm | |
| S$_1$ | 0.0050 | 0.026 | 0.032 | 0.015 |
| S$_2$ | 0.0029 | 0.015 | 0.019 | 0.0064 |
| S$_3$ | 0.0090 | 0.03 | 0.036 | 0.021 |
| S$_4$ | 0.0016 | 0.01 | 0.014 | 0.0043 |
| S$_5$ | 0.0043 | 0.019 | 0.026 | 0.012 |
| S$_6$ | 0.0027 | 0.024 | 0.029 | 0.0048 |
| S$_7$ | 0.0042 | 0.017 | 0.018 | 0.0087 |
| S$_8$ | 0.0020 | 0.009 | 0.013 | 0.0045 |
| S$_9$ | 0.0015 | 0.006 | 0.0088 | 0.0019 |
| S$_{10}$ | 0.0023 | 0.011 | 0.014 | 0.0051 |

Note: Location of the sample is shown in Figure 2 as S$_{number}$.

### 3.1.5. Evidence from Break-Outburst Sediments

The dam break-outburst sediment is also one record of a landslide dammed lake, and it is also an important way to understand the dam-break process, which is usually difficult to find in an old river blocking event. According to the particle size of dam break-outburst sediments, the flood parameters at that time can be obtained by back analysis [22,24]. Furthermore, reasonable analysis of dam break-outburst sediments can also be made to determine the sequence of river blocking events.

In short, for a complete blockage of the river, starting with the blockage of the river by a landslide and ending with a dam failure, we can mainly investigate, describe and summarize from the above five aspects. Among them, many lacustrine deposits is the most critical and convincing evidence for long-term existence of river blocking.

### 3.2. Analysis of Interdependence Landslide Damming of River Events

The characteristic of the study area is that there have been many river blocking events. Therefore, more data are needed to explain whether these river blocking events interact with each other, which may be inconsistent with or even contrary to the results obtained from a single analysis of river blocking. These problems will mainly affect the judgment of river blocking time and thus affect the order of river blocking events, so more means and evidence are needed to explain the overall process of river blocking events. For example, if the WDL dammed lake formed early and lasted for a long time, then the dating age of the lacustrine sediments is likely to indicate WDL rather than other dams upstream. Besides, considering a long time of dammed lake existence, the effect of water on the genesis of other landslide dams shall be considered in numerical simulation even in such dry and rainy areas.

### 3.2.1. Elevation Inference

In terms of elevation, there is a rule that $E_{dam}$ (the elevation at stable formation of the dammed lake) $\geq E_{lake}$ (the highest elevation of the dammed lake) $\geq E_{lacustrine}$ (the highest elevation of the lacustrine sediments). Since the ancient barrier lake has disappeared, we can obtain information from the present shape of the dam and the highest retention elevation of the lacustrine sediments. If the lacustrine sediments and landslide dam belong to the same river blocking event, then the highest elevation of the former cannot be higher than that of the latter. If not, the lacustrine sediments would not be formed by the dam. The principle is to rely on the geological boundary.

For $E_{dam}$, we obtain the profile chart according to the DEM, then the original dam shape is roughly outlined in reference to the form of the Baige landslide which occurred in the upstream according to Feng, et al. [25]. Finally, the reasonable elevation value (Figure 6) is deduced. For $E_{lacustrine}$, we use lacustrine sediments elevation recorded on field investigation by comparing the relatively highest point (Table 2).

**Table 2.** Elevation of dam crest and maximum elevation of lacustrine deposits upstream from the dam.

| Dam | Minimum Elevation of Dam Crest (m) | Maximum Elevation of Lacustrine Deposits Upstream the Dam (m) | Dam | Minimum Elevation of Dam Crest (m) | Maximum Elevation of Lacustrine Deposits Upstream the Dam (m) |
|---|---|---|---|---|---|
| WDL I | 2500 | 2426 | SWL | 2394 | 2430 |
| WDL II | 2463 | | SDX | 2423 | 2445 |
| RCR | 2444 | 2442 | GD | 2455 | 2446 |

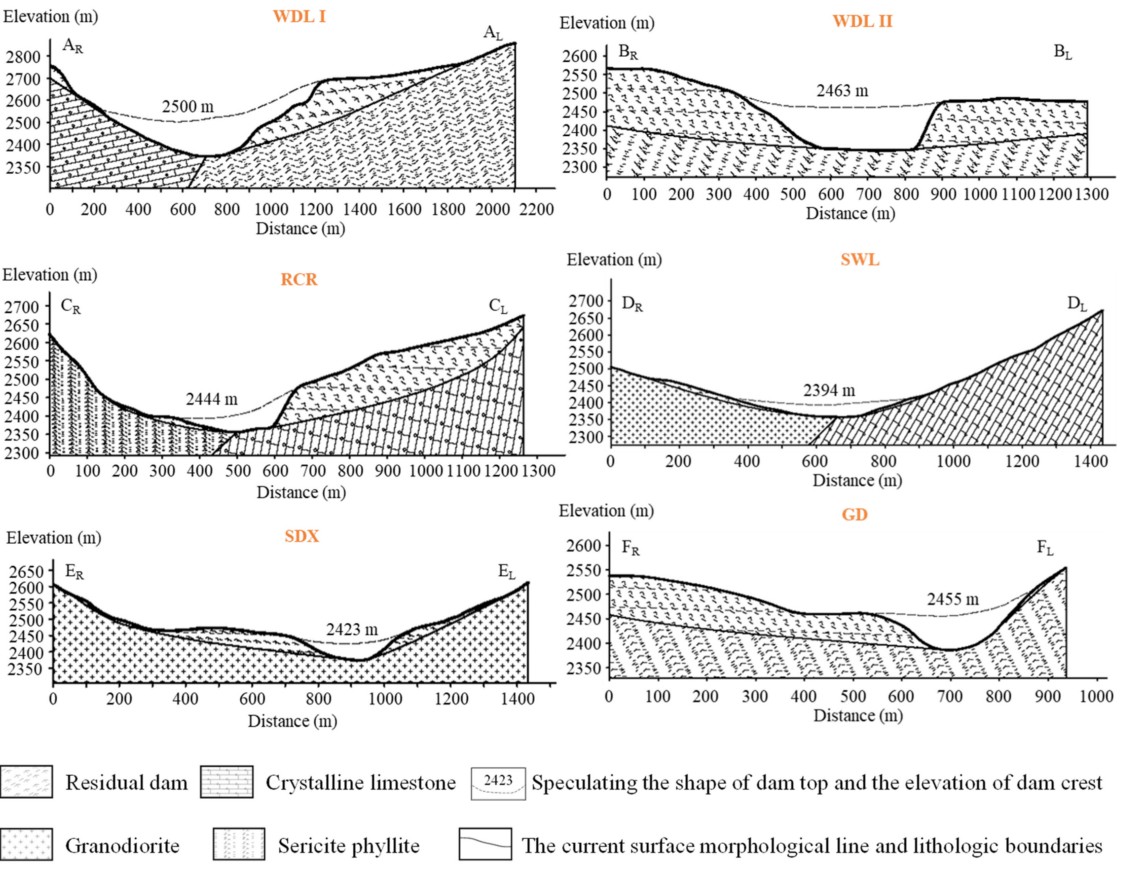

**Figure 6.** Speculative profile of each dam.

### 3.2.2. Dating

For several river blocking events, dating is a direct method to determine the sequence and, especially, multi-method dating campaigns enhance our understanding of the beginning and end of the river blocking event. However, due to the limitations of objective conditions, such as the error of test methods, the lack of availability of dating samples, insufficient funds and the uncertainty of whether the obtained samples have been in the accumulation body or later mixed in, it is often necessary to analyze them from multiple aspects using different methods [26]. Although more direct and high precision evidence is the $^{14}$C dating age of the dam material, $^{14}$C dating requires high wood charcoal samples. First, this section belongs to the dry and hot valley, and there is less vegetation on both sides of the river, so the sample collection is very difficult. Second, the source of samples cannot be guaranteed, so data may be deceptive. Therefore, the dating data of lacustrine sediments can often be used to assist the explanation. At present, the optically stimulated luminescence (OSL) dating method is widely used. In our paper, we commissioned the Institute of Hydrogeology and Environmental Geology to carry out OSL dating of the samples. To enrich the number of samples, we also used sample data from published literature [16,17].

In a single river blocking event, it is reasonable to infer that the normal sequence is formed later in the upper than lower part; in other words, the bottom is older than the top. Through field investigation, there is no sequence inversion caused by tectonic movement. According to the stratigraphic relationships between the dam body and the lacustrine sediments, a relative age for the dam can be concluded.

However, for several river blocking events, due to the influence of river geomorphology (Figure 7), the analysis of the dating results of the lacustrine sediments can be divided into the following situations.

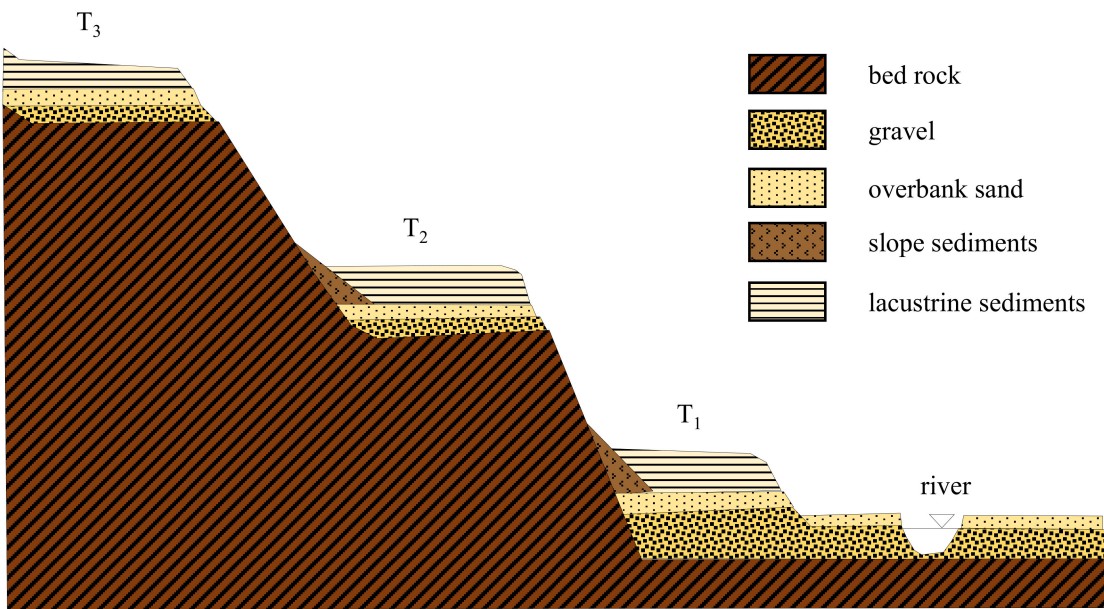

**Figure 7.** Diagrammatic drawing of river geomorphology.

Situation 1 is the formation of lacustrine sediments in the same river blockage with the same bottom baseline (Figure 8a). In general, through the detailed investigation on the site, the relative bottom and the relative top of the lacustrine sediments are found. Through the time difference between the top and bottom, we can roughly infer the duration of the river blocking.

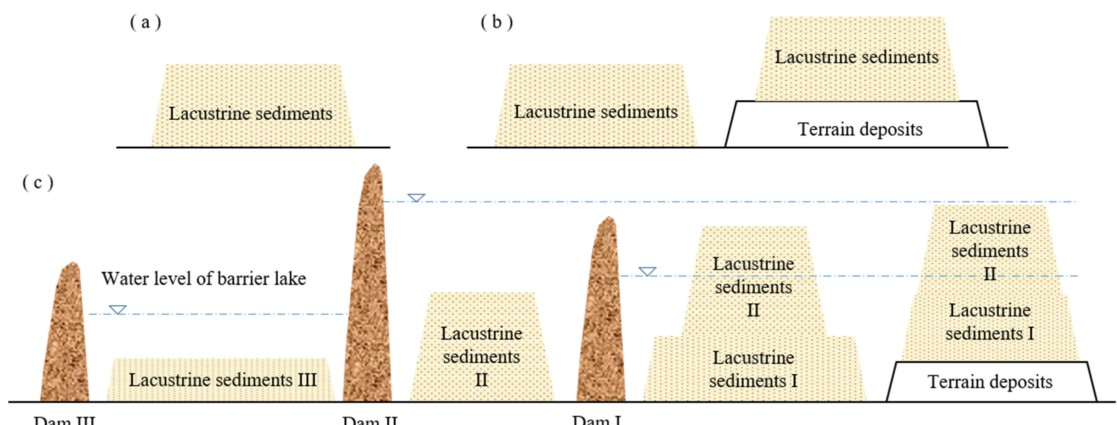

**Figure 8.** Diagram of the relationship between lacustrine sediments. (**a**) Situation 1; (**b**) Situation 2; (**c**) Situation 3 and Situation 4.

Situation 2 is in the same river blockage with the different bottom baseline (Figure 8b). We find the bottom of the profile of the right-hand sampling point then subtract the bottommost elevation of the left-hand sampling point from the bottommost elevation. Finally, we subtract the difference in elevation obtained above from the right-hand sampling point. The same bottom elevation is obtained according to the above method and then analyzed according to situation 1. When the bottom cannot be determined, the difference can be made by using the elevation of sampling points of the same age.

Situation 3 is in different periods of river blocking events with the same bottom baseline (Figure 8c, Dam II with Dam III). The age of sediments in the bottom is most likely different. Data analysis can often form two series.

Situation 4 is in different periods of river blocking events with the different bottom baseline (Figure 8c, Dam I with Dam II), first according to situation 2, then according to situation 3.

These situations only consider the general case, not all. The specific method will be described below.

We first assume that several river blocking events are independent of each other, and then we make the chart (Figure 9) according to dating data. The analysis results of these data are obviously contrary to the assumption preceding part of the text that there is a negative linear correlation between years and elevation. These data are dependent and need to be processed further. According to the results of dating data and its errors, the frequency statistics are carried out with 100-year intervals, and four peaks are found (Figure 10). Under the guidance of no clear experimental purpose, the results of random sampling are related to the distribution of samples, so it can be considered that these dating data roughly represent the four river blocking events.

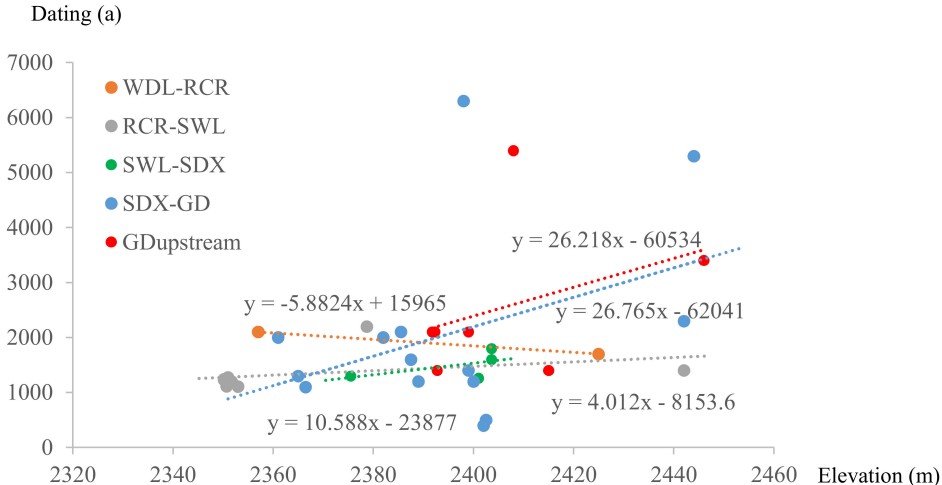

**Figure 9.** Unprocessed OSL dating data.

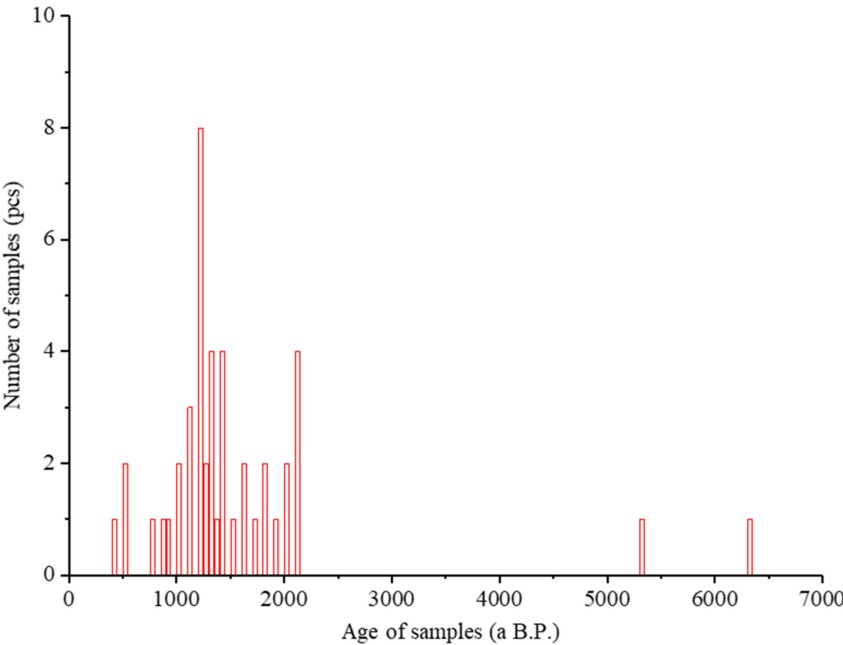

**Figure 10.** Frequency histogram of OSL dating data.

Therefore, we processed the data as follows:

First, the data were classified by river sections according to their locations.

Then, in each category, the classification was further carried out according to the linear relationship.

Last, the classified data were reasonably segmented combined with the results of the age–frequency histogram, and the classification results were processed into the same baseline to obtain the results (Figure 11).

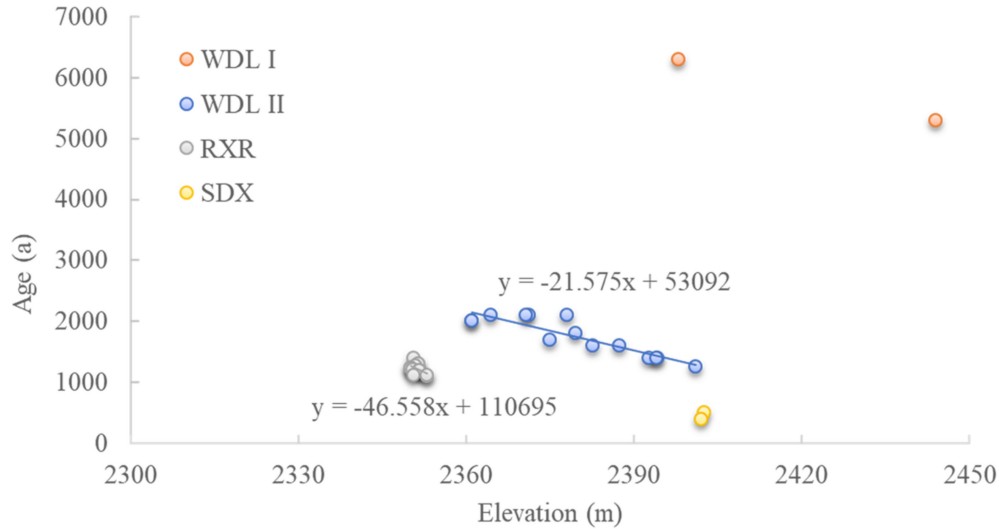

**Figure 11.** Grouping results of post-processing OSL dating data.

### 3.2.3. Interpretation of Geological Phenomena

Good analysis results should be able to reasonably explain the observed phenomenon. We explain the field investigation on this basis to show that the above results are correct.

First, considering landslide dam, GD and SDX residual dam body are relatively complete, the distance between the right bank dam and left bank is short, and the collapse occurs at the cross section, which indicates that the dam body exists for a short time.

Second, we consider the characteristics of dam break-outburst sediments. In the upstream of SWL and SDX, the sedimentary layers under different hydrodynamic environments are found, and the maximum number of accumulation layers in SWL (Figure 12) is more than that in SDX. This is because the formation of SWL is earlier than that of SDX and GD, so it is affected by the two river blocking events.

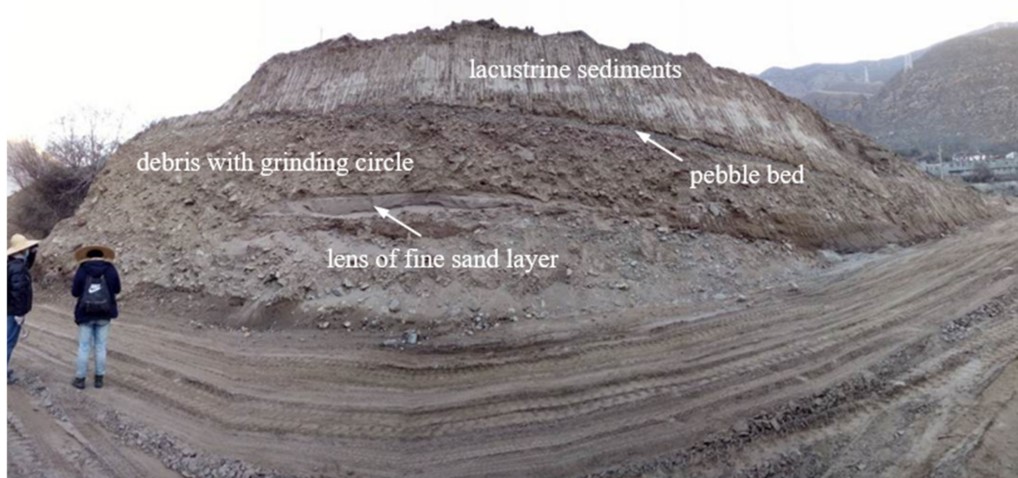

**Figure 12.** Stratification phenomenon in SWL-SDX reach.

Third, from the point of view of lacustrine sediments, the stratification of lacustrine sediments should be more obvious and nearly horizontal in the general long-term stable water environment, while the stratification of lacustrine sediments found in SWL, SDX and GD deposits is not obvious, indicating that the water environment is not stable in the long-term. It was also found that the bedding of the lacustrine sediments was inclined, indicating that the sediments formed before landslides (Figure 13). The horizontal continuous lacustrine sediments in the WDL-SWL reaching up to hundreds of meters are more obvious than that in the SWL-GD reach. Therefore, according to the results of Section 3.2.2, the sequence of each river blocking is relatively reasonable, which is consistent with the results of field investigation.

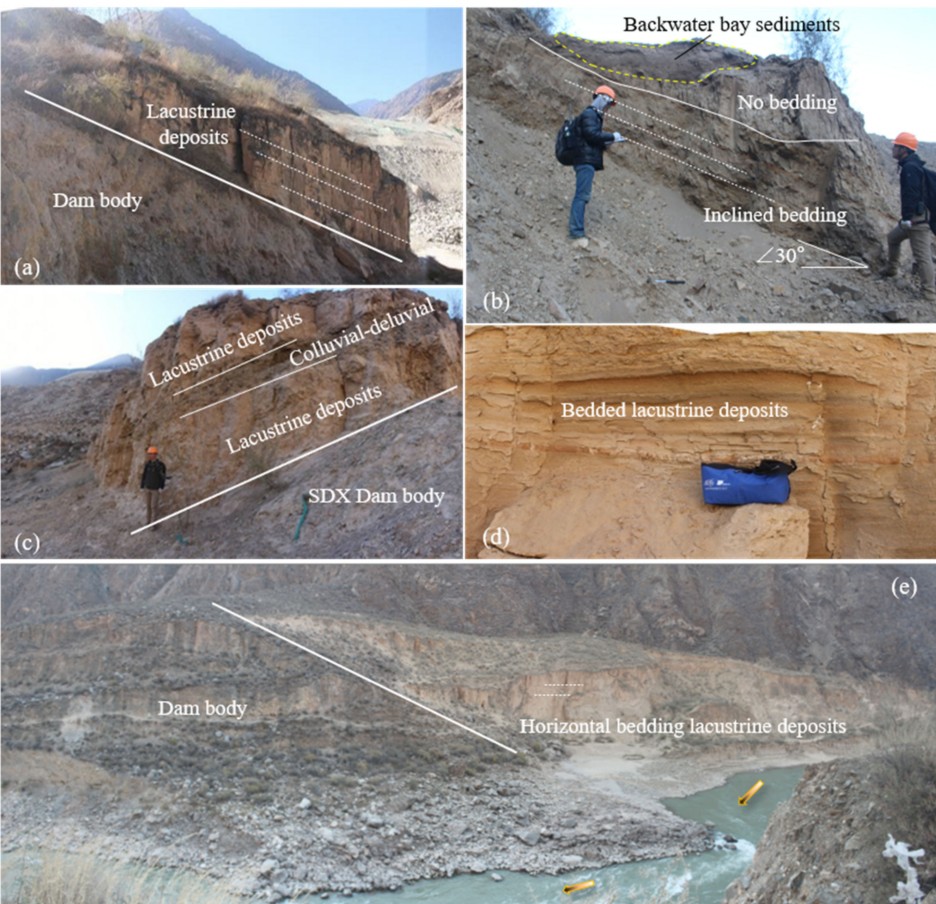

**Figure 13.** Typical geological phenomena. (**a**) Inclined lacustrine sedimentary layer on SDX dam body upstream. (**b**) Horizontal sedimentary layer covers inclined sedimentary layer phenomenon on GD dam body. (**c**) Inclined lacustrine sedimentary layer on SDX dam body downstream. (**d**) Horizontally stratified lacustrine sediments in WDLII-RCR section. (**e**) Horizontally bedding lacustrine deposits layer on RCR dam body upstream.

### 3.2.4. River Long Profile Morphology

Fluvial response causing by landslide dam may theoretically influence sediment yield, channel planform, cross section, gradient, or bed configuration [27]. Among these potential response variables, researchers are interested in long-term fluvial response, especially in channel gradient (Figure 14). By finding the turning point between the gentle gradient and the steep gradient, knickpoints can be recognized in the river long profile [28]. In our paper, the following steps were used to obtain a profile of the river section:

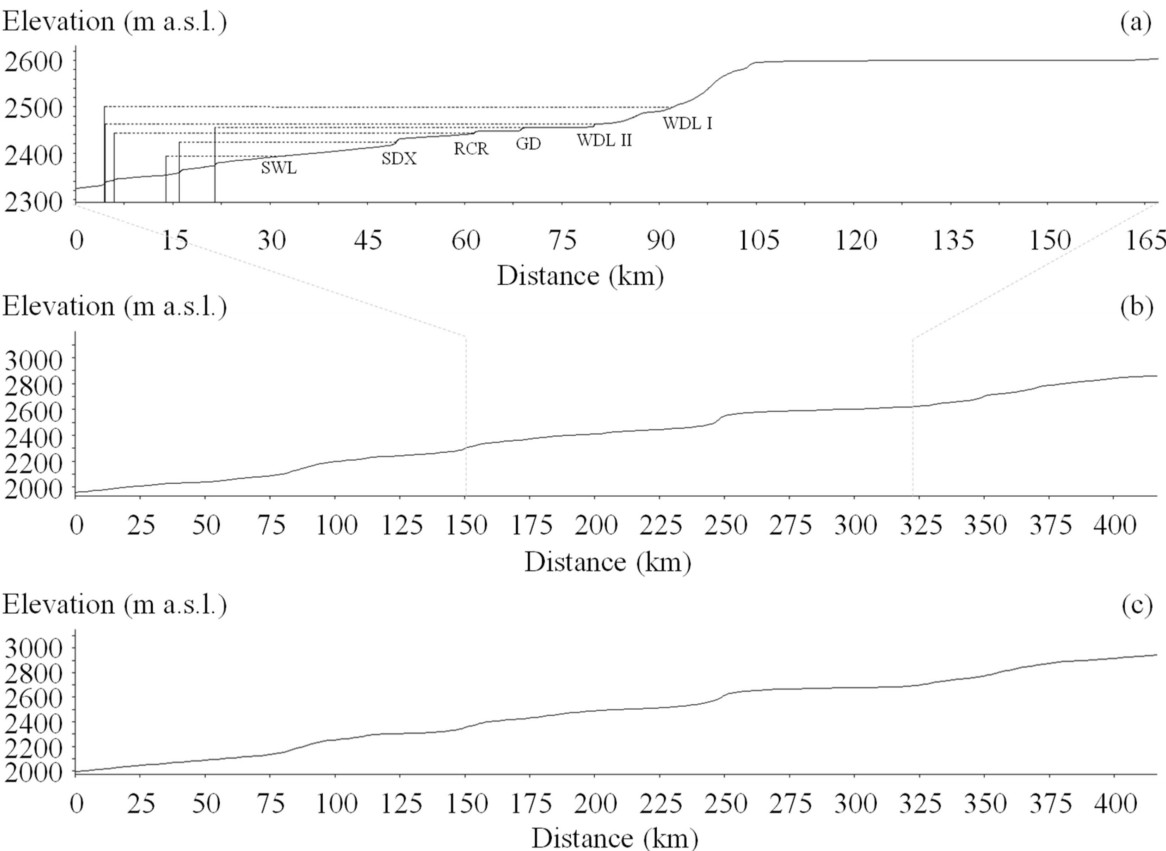

**Figure 14.** Fluvial response to river blocking dam. (**a**) is generated from ALOS 12.5 m DEM. (**b**) is generated from ALOS 12.5 m DEM in order to determine the scope of (**a,c**) which is generated from GDEMV2 30M as the control group of (**b**).

First, generate Jinsha River channel lines; second, generate points every 5 m from the river line; third, extract elevation values to points; fourth, generate a river length profile by using elevation and length; finally, sketch the profile and find the corresponding elevation of the dam. The first four steps were completed by ArcGIS, and the last step was completed by AutoCAD.

3.2.5. Deformation Analysis

Usually, the longer the accumulation body exists, the more tectonic activities it experiences, causing relative instability, vulnerable to erosion, and gradually disappearing. This explains why the longer the time the river blocking event is, the harder we find the remaining dam.

In this paper, the surface deformation of the residual dam body was analyzed through the SARscape module of ENVI 5.3 using remote sensing images from January 2018 to December 2020. Then, the average deformation rate and the response to the flood from Baige barrier lake were obtained of both the whole dam body and part of the whole area along the river (Figure 15).

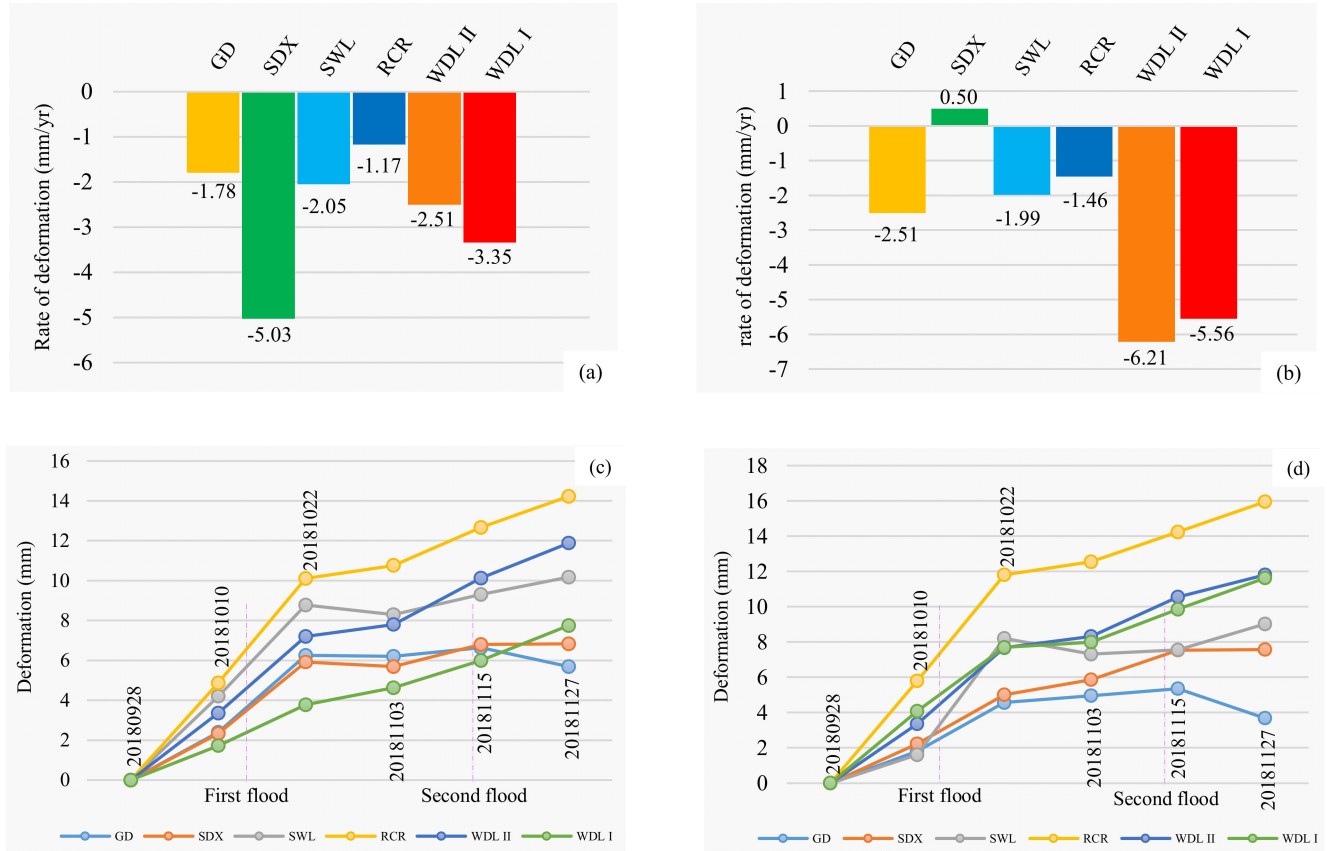

**Figure 15.** The surface deformation of the residual dam body. (**a**) The average deformation rate of the whole dam body; (**b**) the average deformation rate of part of the whole area along the river; (**c**) the whole dam body response to the flood from Baige barrier lake; (**d**) the response to the flood from Baige barrier lake of part of the whole area along the river.

## 4. Results and Discussions

Using the proposed analytical approach of multiple river blockages, we derived the evolutionary sequence of the dams blocking the river in this section. Based on Table 2, we can analyze that WDL I, II and RCR river blocking events have a wide range of influence, and other river blocking events might have been affected. Therefore, the influence of each other must be considered in sampling analysis.

On this basis we analyzed the OSL dating data; the fitted data are shown in Figure 11. Compared with the results of $^{14}$C dating data [17], the results of WDG II are very close, indicating that the analysis method is suitable. The approximate time of river blocking in this reach can be obtained: the river blocking occurred in WDL I reach about 6300 years ago, and the duration of river blocking is about 1000 years; the river blocking occurred in WDL II reach about 1900 years ago, and the duration is about 400–840 years; the river blocking occurred in RCR reach around 1300–1400 years ago, and the duration is about 190–370 years; the river blocking occurred in SWL reach about 1370 ago, and the river blocking was relatively short; at least two landslides occurred in SDX, the first at 750 a B.P. on the right bank and the second at 510 a B.P. on the left bank. The river blocking duration was about 100–110 years, the GD river blocking time was about 900 years ago, and the river blocking duration was uncertain because of fewer data. The above results can explain many implausible geological phenomena.

Finally, we found further evidence corresponding to the deformation of the dam and the long profile of the river. It can be seen from Figure 14 that the river blocking event not only generates knickpoints at the dam site, but also generates knickpoints at the upstream. Moreover, the results of the points analysis are basically consistent with the above dating

data. The blocking age of WDL I and WDL II is long, and the change of river channel is obvious. The SWL blocking event lasted for a small time and had little effect on the river. In terms of elevation, the height fitted by the profile shape of the dam body seems to be conservative, such that the height of the WDL I may initially reach 2600 m.

By analyzing Figure 15a,b, we can draw a conclusion that the dam bodies are in denudation state, but the denudation rate is small, which is consistent with the situation that the accumulation dam can exist for thousands of years. Comparing GD, SWL, WDL I and WDL II in Figure 15a, the older the age, the greater the deformation rate; SDX is abnormal in Figure 15a because the construction camp of Suwalong Hydropower Station is built on it, and it has been artificially transformed, while in Figure 15b of the control, the part of the whole dam along the river is slightly uplifted mainly due to the uplift of the region with around 5 mm/yr rate [4]. The RCR river section is relatively small because this reach may be a relatively "sedimentary area" in the whole study reach. As can be seen from Figure 15c,d, flood is not the main reason for the disappearance of the dam. After the flood, the dam body is in a state of accumulation rather than erosion, and the accumulation thickness of RCR is the largest in the whole river section of the study area, which may be related to the existence of a large number of continuous lacustrine sediments in the upstream of RCR. Moreover, comparing Figure 15a,b, it can be concluded that the erosion rate along the river section of RCR is lower than the whole erosion rate, because the material on the upper part of the dam body is eroded and stripped from the original position and then deposited in the lower part of the dam body along the river.

If the above deformation can only explain the result of a short time, the landform of the accumulation dam body is the result of long-term evolution. Through the rough measurement of two relatively distant points on both sides of the residual dam body by Google Earth, the erosion section length perpendicular to the river of WDL I is about 1100 m, WDL II is about 500 m, RCR is about 350 m, SDX is about 190 m, and GD is about 200 m; these data are positively correlated with the ages of the accumulation dams.

The paper also summarizes a systematic process for investigating river blocking incidents. The geological environment of the research area breeds a wide range of mass movement, which creates good conditions for the occurrence of the river blocking event. For an ancient river blocking event, whether dammed lakes persist for a short time or a long time [29], there will always be some traces near the river block. Five important aspects of the field investigation of the river blocking event need to be paid attention to. Among this evidence, the dam body on both sides is the most intuitive evidence of ancient river blocking event, because it is one of the necessary conditions for the formation of the river blocking; and the wide range of lacustrine sediments accumulated behind the dam is convincing and abundant evidence (because its existence often indicates longer duration of dammed lake). For example, based on lacustrine sediments located in the same profile, we could roughly predict the age and duration of the river blocking event by calculating the difference between the age of the top layer and the bottom layer; then, we could obtain the average deposition rate by dividing the difference by the thickness of the lacustrine sediments, which is related to the sediment content, fine particle composition and climate; the mineral composition of the sediment; and chemical composition can reveal the source of material and sedimentary environment, etc. In addition, because of its widespread existence and fine-grained sediment with low cementation strength, it makes sampling convenient. Compared with the $^{14}$C dating (samples need higher requirements and are not easy to obtain; sometimes we are even unable to find the appropriate test sample), the samples used for OSL dating of lacustrine sediments are easily found and obtained, which can increase efficiency and prevention of accidental errors. However, when multiple river blocking events occur in a river reach interacting with each other, if the sample is not carefully distinguished from which river blocking event, the analysis results from OSL dating will be seriously affected. In this paper, a feasible method for analyzing several river blocking events is proposed. Then, combined with the geological phenomena of field investigation, the relative reasonable sequence of river blocking is obtained, and the age

of some river blocking events that previous researchers did not give clear results for have also been identified. An accurate activity history is of great significance not only for the further study of the tectonic activity and sedimentary climate at that time, but also for the numerical simulation of dam-break and accurate back analysis of the event of landslide blocking river occurrence.

At the same time, our team finds that the error of OSL dating by using lacustrine sediments is within the acceptable range, but the following proposals must be paid attention to: (1) Before sampling, the experimental plan must be made and the samples should be collected purposefully. (2) Light should be avoided when sampling, and timely experiments should be conducted after sampling. Otherwise, the sample needs to be sealed, avoiding light, and stored at room temperature. (3) The determination of moisture content of samples is a very important error factor, which is not only the current moisture content, but also the average moisture content of samples in the historical process. (4) Under money- and time-permitting conditions, parallel samples should be collected around the sample and a profile sequence should be collected for dating in order to reduce accidental errors and improve the accuracy of the results.

Although the research method and research ideas are quite complete, the research is limited because of the complicated geological environment of the reach, the urgent time in the investigation, the inaccessibility of some positions to carry out the field investigation, and the inability to take samples due to the high sampling location. Therefore, the results presented are one of the development processes of river blocking with a greater probability. More discussion is welcome from future researchers. Finally, in order to make the relevant investigation more detailed and efficient, we made a systematic summary based on our team and previous studies, and put forward the following complete investigation and analysis flow chart (Figure 16), so that the relevant investigation can be more scientific and comprehensive, and fully reflect the relevant geological and geomorphic information.

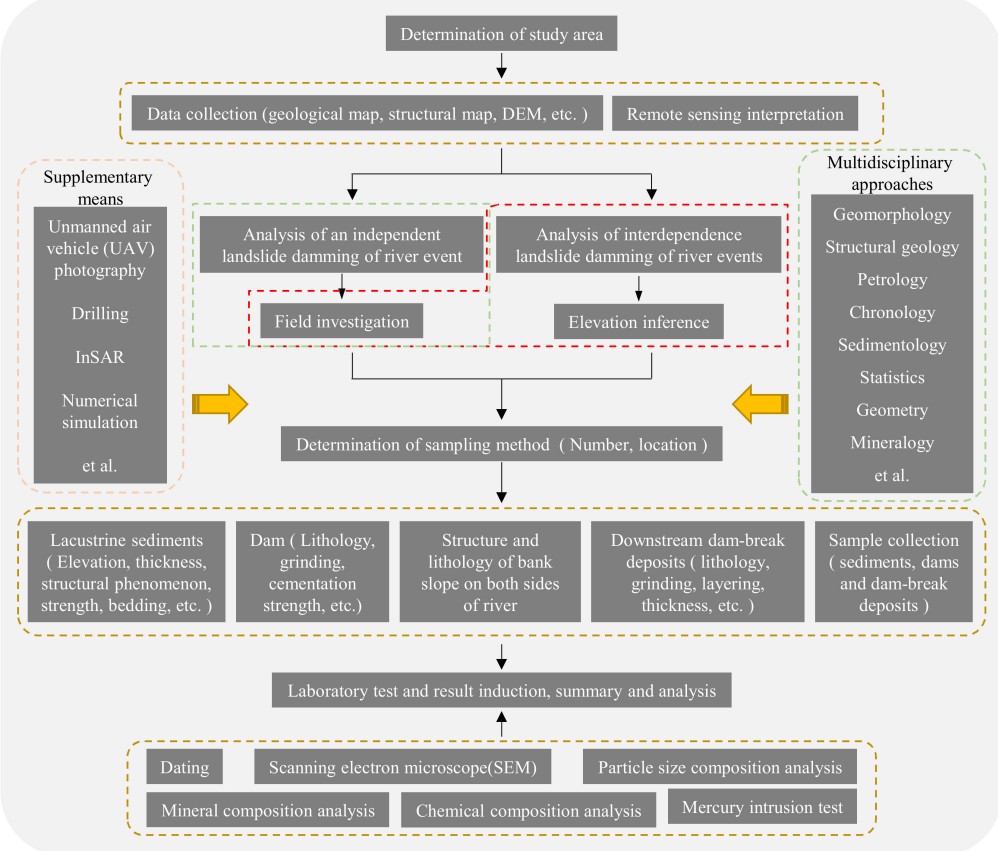

**Figure 16.** The general flow chart of studying the river blocking event.

## 5. Conclusions

There are very few studies on the age determination of dams with multiple river blockages. This paper presented an applicable method for investigating river blocking bodies and studying multiple river blocking processes in the same reach. Through this method, the relatively reliable river blocking sequence evolution history of Wangdalong-Gangda reach was scientifically and effectively restored. The results of elevation and dating analysis showed that the river blocking occurred in WDL I reach about 6300 years ago, WDL II reach about 1900 years ago, RCR reach around 1300–1400 years ago, SWL reach about 1370 ago, SDX reach about 750 years ago (a time away from today) and 510 (a time close to today), and the GD river blocking was about 900 years ago. The formation of these barrier lakes led to the decrease of river dynamics and river deposition effect, which may be an important reason for the inhibition of river channel incision and may directly affect the evolution of the local landscape [30]. From this perspective, the above results were also well verified by geological phenomena, river long profile morphology and accumulation deformation. This study is of great significance because it shows that the integrated method can provide a reasonable explanation for the evolution of the history of river closure in this area and provide a reference for future related research.

**Author Contributions:** Conceptualization, Y.Z., J.C. and Q.W.; methodology, Y.Z. and Y.L.; investigation, Y.Z., J.C., Y.L., F.G. and C.C.; Validation, J.C.; data curation, S.S.; writing—original draft preparation, Y.Z.; writing—review and editing, J.C.; visualization, Y.Z.; supervision, Q.W.; project administration, J.C.; funding acquisition, J.C., Q.W. and S.S. All authors have read and agreed to the published version of the manuscript.

**Funding:** This research was financially supported by the Key Project of NSFC-Yunnan Joint Fund (Grant no. U1702241), the National Natural Science Foundation of China (Grant No.42177139) and the National Key Research and Development Plan (Grant No. 2018YFC1505301).

**Institutional Review Board Statement:** Not applicable.

**Informed Consent Statement:** Not applicable.

**Data Availability Statement:** Not applicable.

**Acknowledgments:** The authors would like to thank Yuchao Li, Zhihai Li, Zhaoxi Wang, Jiejie Shen et al. for their contributions, comments and suggestions which helped in making this paper better.

**Conflicts of Interest:** The authors declare no conflict of interest.

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
