# Peer review of "Sequence Analysis of Ancient River Blocking Events in SE Tibetan Plateau Using Multidisciplinary Approaches"

_water, doi:10.3390/w14060968_

Round 1
Reviewer 1 Report
The manuscript “Sequence Analysis of Several Ancient River Blocking Events in the Suwalong Reach of the Upper Jinsha River Based on Multidisciplinary Approaches, SE Tibetan Plateau” is devoted to a very important topic: the research of river blocking events and their dating problems. The research is based on multidisciplinary method which provides a lot of valuable data to substantiate the authors’ conclusions. However there are some major flaws in the paper and also quite a lot of grammar mistakes and typos. You can find major remarks below:
- The overall merit of this research remains unclear. Why did the authors decide to conduct such kind of research? Were there any previous attempts to define the age of those dams or consequences of those blocking events? If there were other studies of this region are the results presented by authors consistent with previous findings or not?
- Why were samples taken? What was done with them afterwards? This information is not present in the article.
- The Methodology section is too long and contains a lot of the research results. Please re-arrange sections. Some valuable information obtained by the authors should be moved to Results and Discussion section. Leave only methodology (basics of multidisciplinary method that was used, the information about sampling process, how was the sediment dating done, etc.) and equipment description here. All specific results with corresponding figures should be moved to the Results and Discussion section.
- Please consider to shorten goal three in Introduction it is too long now. What was the overall aim of this research? Why is it so important?
- The goal of this research has been already defined in Introduction section. Why there is some other research purpose in the methods section? Please check and re-phrase.
- How were the maps of Figures 2 and 3 created? What software was used for it? Please add this information in Methods section.
- Did the authors study the lithology of the residual dams and bank slopes?
- It is not clear how did authors obtain density of sediment particles parameter value for equation 1 (rS = 2700)?
- How were lacustrine sediments distinguished from river sediments? On what basis?
- How was the grain size measured? What equipment was used?
- Dating method described in 3.2.2 section is a bit confusing. It is not clear how did the authors date the sediments after all? Was their results compared with literature data of 14C-dating or not? Please re-write this section and divide methods from results. Results should go to Results section.
- Methods of fluvial response to River blocking dam (Fig. 14) generation were not defined. What software and what data were used? Please clarify.
There are also several minor remarks:
Please check the labels for faults in Figure 1 - some are written with a capital letters, and some with lowercase.
Line 114: the combination of modern scientific and technological is applied -> “scientific and technological” what exactly? These are adjectives and associated noun is missing.
Lines 119-120: My 119 main research purpose -> Please rephrase. Firstly a scientific article should not contain personal pronouns. And secondly this research has more than one author.
What are thin red lines on Figure 2?
What is “I26 m” on Figure 5a?
Line 217: a long time existing dammed lake -> Did you mean “a long time of dammed lake existence”?
Lines 224-225: that the present shape of the dam and the highest retention elevation of the lacustrine sediments -> that the present shape and the highest retention elevation what? The verb is missing.
Line 232: Elacustrine -> “lacustrine” should be in subscript.
Line 244: river blocking event However -> dot is missing.
Lines 263, 268, 271, 274: Situations 1, 2, 3, 4 -> Situation 1 and so on (singular noun).
Lines 269-270: please rephrase the sentence; it is hard to understand in present form.
Line 273: most likely difference -> different.
Figure 9: what are the units of Y axis?
Figure 11: there are no scatter plot visible (no points on the plot). Please check.
Figure 16: (c) the response to the flood from Baige barrier lake of from both the whole dam body -> please rephrase.
Line 423: (sample -> closing bracket is missing.
Lines 427-430: this sentens is confusing, please clarify.
Figure 16 looks like appendix.
However I believe that this research lies within the scope of Water Journal and will make a good contribution to it after significant revisions. The manuscript can be accepted after major revisions.
Author Response
Thank you for your efforts in reviewing our manuscript and we hope that you will be satisfied with our revisions. Detailed responses are in this document named “cover letter of author comment”. If you have any questions please contact us.

Reviewer 2 Report
Dear Authors,
Please refer to the revisions/suggestions mentioned in the text.
Regards

Author Response
Thank you for your efforts in reviewing our manuscript and we hope that you will be satisfied with our revisions. Detailed responses are in this document named “Cover letter of author comment”. If you have any questions please contact us.

Round 2
Reviewer 1 Report
Authors did a very good job in improving the manuscript “Sequence Analysis of Several Ancient River Blocking Events in the Suwalong Reach of the Upper Jinsha River Based on Multidisciplinary Approaches, SE Tibetan Plateau. However there are still some language problems (see some examples below) and a lot of valuable information from authors’ answers (cover letter) was not included in the text of the paper. Firstly, I strongly believe that the same questions could arise from the other readers. So could you please include the key points of your answers in the manuscript itself?
Secondly, there should be information in the Methods section describing briefly all laboratory techniques that authors did use in their research. For example, I did not find any additional information on the laser particle size analyzer and hydrostatic sedimentation experiment in the new version of the manuscript. I suggest adding all valuable data about methods either in separate section or in the corresponding sections (e.g. data from lines 148-149 of the cover letter could be placed at section 3.1.4 of Methods, where authors use these results). As for OSL dating, some more information about this analysis incorporating authors’ response (some data from lines 155-160 of the cover letter) can be added to section 3.2.2., for example. Information from lines 173-179 of the cover letter could be placed in section 3.2.4 (In brief). And so on.
There are also several minor remarks:
Line 53-54: It is more interested in how we can learn right the ancient information from these events. -> “It is more interested in how” doesn't make sense in this sentence. Who is this “it”, who is not interested? Please, check.
Lines 69-70: ideas about analysismultiple river blockages -> analysis of multiple river blockages?
The manuscript can be accepted after major revisions.
Author Response
Thanks for your efforts to improve our manuscripts. Please check the Cover letter of author comment for the changes you have suggested. More specific details are in the uploaded revised manuscript.

Round 3
Reviewer 1 Report
Authors have included some valuable information from cover letter to the text of the paper itself. The Manuscript has become clearer and can be published in present form.